

# Here are the polyps: in situ observations of *jellyfish polyps and podocysts* on bivalve shells

Lodewijk van Walraven[1], Judith van Bleijswijk[2] and Henk W. van der Veer[1]

[1] Department of Coastal Systems, and Utrecht University, NIOZ Royal Netherlands Institute for Sea Research, Den Burg, Netherlands

[2] Department of Marine Microbiology and Biogeochemistry, and Utrecht University, NIOZ Royal Netherlands Institute for Sea Research, Den Burg, Netherlands

## ABSTRACT

Most Scyphozoan jellyfish species have a metagenic life cycle involving a benthic, asexually reproducing polyp stage and a sexually reproducing medusa stage. Medusae can be large and conspicuous and most can be identified using morphological characteristics. Polyps on the other hand are small, live a cryptic life attached to hard substrates and often are difficult or impossible to distinguish based on morphology alone. Consequently, for many species the polyp stage has not been identified in the natural environment. We inspected hard substrates in various habitats for the presence of Scyphozoan polyps. Three polyps were found on Dogger Bank, Central North Sea, attached to the inside of the umbo of empty valves of the bivalves *Mactra stultorum* and *Spisula subtruncata*. One polyp was accompanied by four podocysts. With this knowledge, the inside of bivalve shells washed ashore in Oostende (Belgium) was inspected and supposed podocysts on the inside of empty valves of *Cerastoderma edule* and *Spisula elliptica* were found. Polyps and podocysts were identified to species level by 18S rDNA and mitochondrial COI sequencing. The three polyps found on Dogger Bank all belonged to the compass jellyfish *Chrysaora hysoscella*. One podocyst from the Oostende beach also belonged to this species but another podocyst belonged to *Cyanea lamarckii*. These are the first in situ observations of *C. hysoscella* and *C. lamarckii* polyps and podocysts in the natural environment. *Mactra, Cerastoderma and Spisula species* are abundant in many North Sea regions and empty bivalve shells could provide an abundant settling substrate for jellyfish polyps in the North Sea and other areas. Several new strategies to increase the detection of polyps on bivalve shells are presented.

## INTRODUCTION

Most Scyphozoan jellyfish species have a metagenic life cycle involving sexually reproducing pelagic medusae producing planula larvae which settle into benthic polyps called scyphistomae, or podocysts (Fig. 1). Scyphistomae (hereafter called "polyps") can reproduce asexually in several ways including by strobilation, releasing juvenile jellyfish called ephyra into the water column (*Adler & Jarms, 2009*; *Arai, 1997*; *Helm, 2018*;

Corresponding author
Lodewijk van Walraven,
lodewijk.van.walraven@nioz.nl,
lodewijkvanwalraven@gmail.com

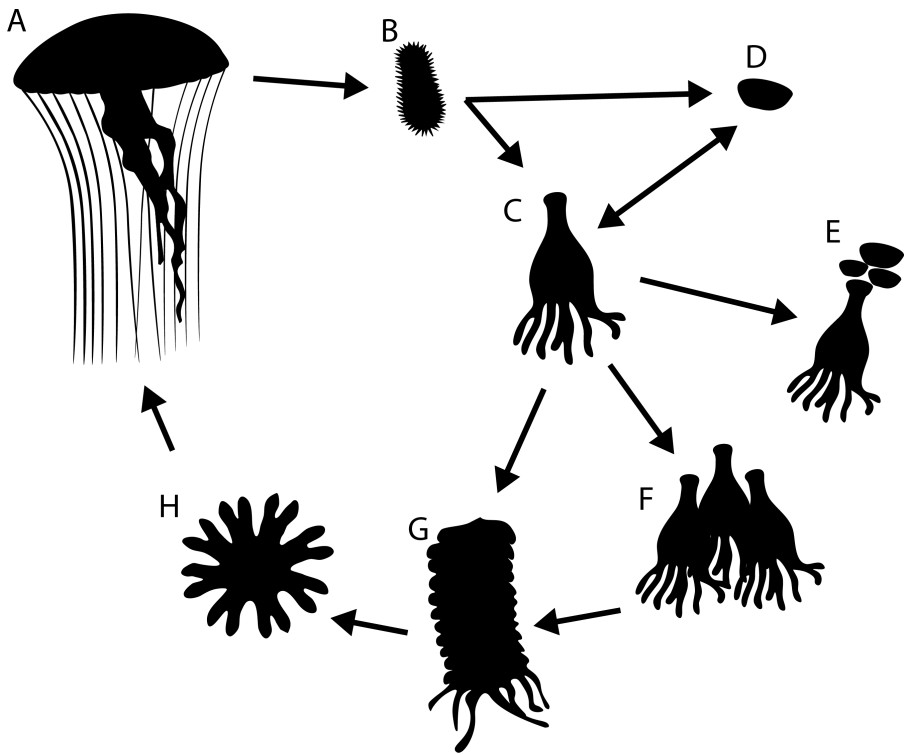

**Figure 1** **Generalised life cycle of metagenic scyphozoa.** Adult medusae (A) reproduce sexually, producing planula larvae (B) which settle on a substrate and either develop into a scyphistomae (C) or directly into a podocyst (D). Scyphistomae can reproduce asexually by producing podocysts (E), producing scyphistomae (F) or forming strobila (G) which release ephyrae (H) in a process called strobilation. Ephyrae develop into adult medusae.

*Schiariti et al., 2014*). Most species can also form podocysts, either directly from planula larvae or beneath polyp pedal discs. Podocysts have a chitinous outer layer and contain reserves which allow them to survive for longer periods in adverse conditions such as food scarcity or high predation pressure by nudibranchs (*Arai, 2009*; *Boero et al., 2008*; *Holst, 2012*; *Lucas, Graham & Widmer, 2012*).

Polyps and podocysts are small, live a cryptic life attached to hard substrates and are difficult or impossible to distinguish based on their morphology alone. Consequently, despite the ecological importance of the polyp stage as the source of metagenic scyphozoan blooms (*Arai, 1997*; *Boero et al., 2008*; *Helm, 2018*; *Lucas, Graham & Widmer, 2012*; *Schiariti et al., 2014*), for many species the polyp stage has not been observed in the natural environment (*Van Walraven et al., 2016*).

In Japanese waters polyps have been observed *in situ* for *Aurelia* species on artificial substrates (*Makabe et al., 2014*; *Shibata et al., 2015*) and *Chrysaora pacifica* polyps and podocysts have been found on shells and stones in dredge samples (*Toyokawa, 2011*).

For the North Sea area, it has long been suggested that benthic stages of scyphozoan jellyfish should occur in the area, based on the observation of ephyrae (*Van der Baan, 1980*). In experiments, the five species occurring in the North Sea area, *Aurelia aurita,*

*Cyanea capillata*, *Cyanea lamarckii*, *Chrysaora hysoscella* and *Rhizostoma octopus* have all been shown to produce planulae that settle on hard substrates and develop into polyps or podocysts (*Holst, 2012*; *Holst & Jarms, 2007*; *Holst & Laakmann, 2014*).

In a previous study, polyps were collected from natural and artificial substrates at various inshore and offshore locations in the North Sea and Sweden and could be identified to species level by sequencing both a fragment of 18S rDNA and a fragment of mitochondrial COI (*Van Walraven et al., 2016*). In this study we continue our search for Scyphozoan benthic stages. We describe two promising, widely applicable approaches to find and identify polyps and podocysts on bivalve shells, which lead to the first *in situ* observation of benthic stages of *Cyanea lamarckii* and *Chrysaora hysoscella* are documented.

## MATERIALS & METHODS

### Sampling

Benthic stages of jellyfish were sampled using two different methods. On a transect from Texel, the Netherlands to Fladen Grounds, Northern North Sea (Fig. 2), the Deep Digging Dredge ("Triple D dredge") was used to sample benthic fauna. The triple D dredge uses a 20 cm wide knife that cuts through the sediment to a depth of 20 cm over a distance of about 100 m, sampling an area of 20 m$^2$ (*Bergman & Van Santbrink, 1994*; *Witbaard et al., 2013*). Sampling in UK waters was approved by the Maritime Policy Unit (Legal Directorate) of the Foreign and Commonwealth Office (ref 33/2018). Subsamples of the "bycatch" from the triple D samples; stones and empty bivalve shells, were collected and stored in seawater at 4 °C. The stones and shells were inspected for the presence of scyphozoan benthic stages by visually examining the substrates submerged in seawater, illuminated with a cold light source. When scyphozoan benthic stages were found on Dogger Bank (55.1711, 3.1626) their presence was confirmed by inspection with a stereo microscope. Scyphozoan polyps were removed from the substrate using the tip of a flame-sterilised scalpel and stored in 95% EtOH in 2 ml Eppendorf vials prior to analysis.

Following the observation of podocysts on empty shells in the Dogger Bank samples, the question arose whether these podocysts would remain attached to the shells when they wash up on beaches. On 19-09-2018, empty bivalve shells were collected along the high-water line of Oostende beach, Belgium (51.2338, 2.9121). Shells were rinsed in seawater and were inspected by eye for the presence of scyphozoan podocysts. In the lab, shells were inspected under a stereomicroscope and podocysts were removed from the substrate using the tip of a flame-sterilised scalpel dipped in 95% EtOH to moisten the podocysts prior to removal, as it was found that dried podocysts would disintegrate or disperse when attempting to remove them.

Podocyst- or polyp containing shells were photographed submerged using a Pentax K5 DSLR with Pentax D-FA 100 mm macro lens. Polyps and cysts were photographed using an Olympus stereomicroscope at 20x magnification using a Bresser Microkular ocular tube camera. Stereomicroscope photographs were used to measure polyps and podocysts in ImageJ (*Schneider, Rasband & Eliceiri, 2012*). Of each polyp/podocyst, the diameter was measured along the horizontal as well as the vertical axis.
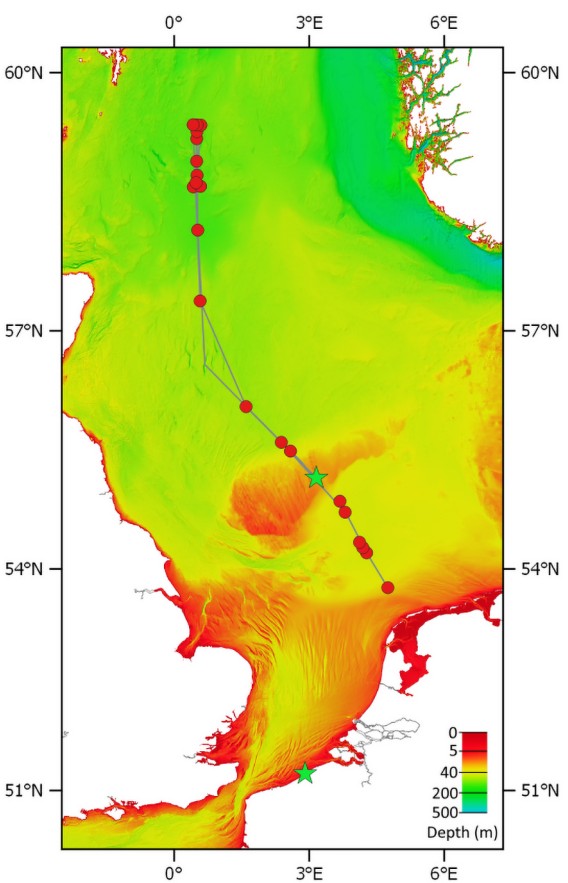

**Figure 2** **Transect of RV Pelagia cruise 64PE438.** Red circles are sampling stations. The green stars show the location where polyps/podocysts were found: on the Doggerbank in the North Sea and on the beach of Oostende along the Belgian coast. Bathymetry layer provided by EMODnet (*EMODnet Bathymetry Consortium, 2016*).

## Genetic analysis

In a dedicated DNA extraction (pre-PCR) lab, polyps and podocysts were transferred to a sterile Eppendorf vial with 200 μl ethanol using a 1ml micropipette (large diameter of opening). After a short spin, most of the ethanol was removed with a 100 μl micropipette, leaving about 20 μl containing the polyps or podocysts. These were washed two times with 1ml of PCR grade water and resuspended in 20 μl lysis buffer taken from the GenElute Mammalian Genomic DNA kit (Sigma). Samples were ground in the Eppendorf vial using a sterile plastic pestle (VWR, 431-0094). Subsequently, DNA isolation was carried out according to the manufacturer of the kit, involving an overnight lyses step at 53 °C and a final elution of the DNA from the silica column in 100 μl Tris buffer (10 μM).

DNA concentrations in the extracts were generally too low to be measured with a Qubit fluorimeter. Exceptions were polyp 1 and 2 and podocyst 7 with concentrations of 0.14, 0.09 and 0.04 ng/ul respectively corresponding to yields between 140 and 40 ng. In 50 μl PCRs containing 2 μl DNA extract, 1 unit of BiothermPlus polymerase (Genecraft), 1x

buffer, 200 μM dNTPs, and 0.5 μM of each primer, diagnostic fragments of 18S rRNA and CO1 were amplified. For 18S rRNA we used primers EUK_F566 (5′-CAG CAG CCG CGG TAA TTC C-3′) and EUK_R1200 (5′-CCC GTG TTG AGT CAA ATT AAG C-3′) with an annealing temperature of 60 °C in 35 cycles amplifying 650 nt of the V4 and V5 region (*Hadziavdic et al., 2014*); for COI we used the jellyfish specific primers FFDL (5′-TTT CAA CTA ACC AYA AAG AYA TWG G-3′) and FRDL2 (5′-TAN ACT TCW GGR TGN CCR AAG AAT CA-3′) with an annealing temperature of 51 °C in 39 cycles amplifying 709nt of COI (*Armani et al., 2013*). PCR protocols involved 2 minutes initial denaturation, followed by 35 or 39 cycles of 45 seconds denaturation at 95 °C, 60 seconds annealing at the temperatures indicated above, and 60 seconds extension at 72 °C. After a prolonged extension of 7 minutes at 72 °C, and a cooling of 4 minutes at 4 °C, the programme ended with a pause at 15 °C. Negative controls did not generate a band. PCR products were Sanger sequenced by BaseClear (Leiden) with forward and reverse primers, and consensus sequences were composed with Chromas Pro software.

Consensus sequences were submitted to GenBank and are available under accession numbers MT074030- MT074034 (COI) and MT075505-MT075509 (18S). The sequences were compared against the NCBI database Nucleotide collection (nr/nt) using BLASTn as a first identification step. Reference trees were built in ARB (*Ludwig et al., 2004*) using the RAxML tool version 8.2.11 (*Stamatakis, 2014*) with sequences from jellyfish species available in GenBank. Newly obtained polyp and podocyst sequences were aligned with Fastaligner, and added to the reference tree using the ARB parsimony option.

## RESULTS

On Dogger Bank, polyps were found on shells of *Spisula subtruncata* and *Mactra stultorum*. On two *S. subtruncata* shells, a single polyp was found. On a *M. stultorum* shell, two polyps were found, with a calyx diameter of 0.31 and 0.90 mm, the latter was accompanied by a cluster of four podocysts (Fig. 3). Podocysts had diameters of 0.34, 0.29 and 0.31 mm (average of 0.31), where one could not be measured as it was obscured by the polyp. At Oostende beach, jellyfish podocysts were found on 23 shells of 7 species: *Spisula subtruncata, Spisula solida, Spisula elliptica, Aequepecten opercularis, Mimachlamys varia, Donax vittatus, Acanthocardia echinata* and *Cerastoderma edule* (Fig. 4). Both cysts and polyps were most often observed in or near the umbo of the shell, always on the inside.

Podocysts were generally circular in shape, with a smooth edge. Podocysts had a central depression, which had an even surface in some podocysts, but in others contained a circular hole, suggesting the podocyst was empty. Minimum podocyst diameter was 0.15 mm, maximum podocyst diameter was 0.61 mm, and average cyst diameter ($n = 71$) was 0.31 mm.

Polyps and podocysts were used entirely for genetic analyses. We were able to obtain amplicons from eleven specimens with the universal eukaryote 18S rRNA gene primers, and amplicons from nine specimens with the COI primers. All amplicons ($n = 20$) were sent off for Sanger sequencing. Results with the universal 18S PCR yielded seven successful identifications: Five specimens came out as Scyphozoa: three polyp and one podocyst were
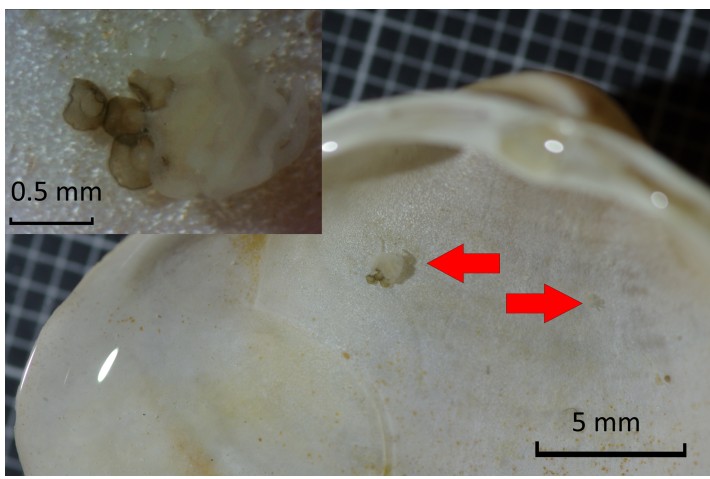

**Figure 3** Detail of *Mactra stultorum* shell with attached *Chrysaora hysoscella* polyps and podocysts.

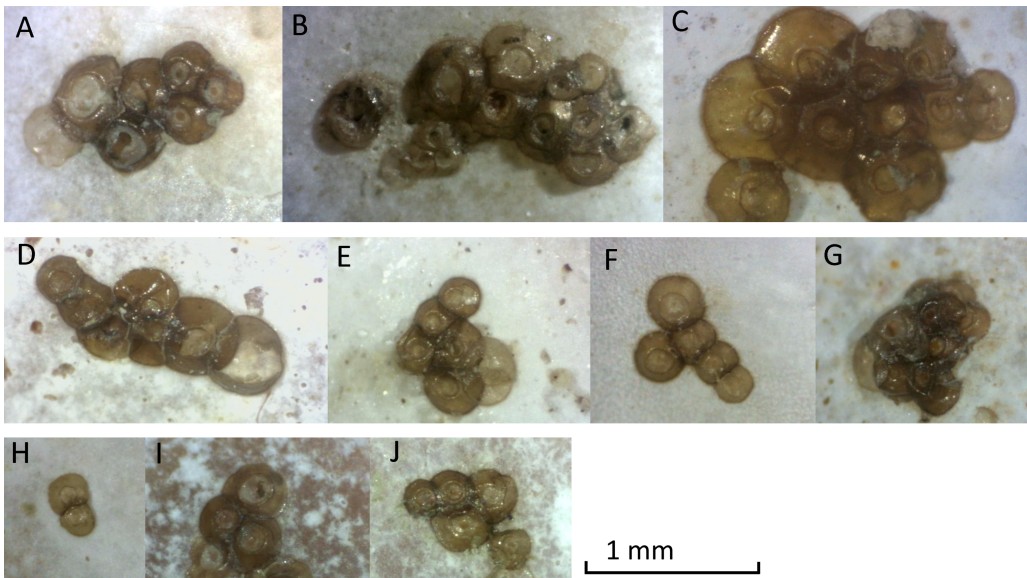

**Figure 4** Photographs of podocyst clusters sampled for genetic analysis. Several podocysts appeared empty, containing a circular hole in the centre, found in clusters A, E, G and J. Cluster G was identified as *Cyanea lamarckii*, cluster I was identified as *Chrysaora hysoscella*. All other clusters could not be identified by genetic analysis.

identified as *Chrysaora hysoscella* and one additional podocyst was identified as *Cyanea lamarkii* (Table 1 and Fig. 5). Two podocysts came out as non Scyphozoa: one as the fungus *Phoma* sp. and the other as the Rhodophyte *Pyropia* sp. All other sequences were not readable. Independent results with COI primers confirmed the identifications of the five Scyphozoa specimens (Fig. 6). One presumed podocyst showed a low similarity (80%

**Table 1  Overview of samples from the two locations in the North Sea area included in genetic analyses.**

| Sample name | Stage | location | date | host shell | ID |
|---|---|---|---|---|---|
| NICO10_p1 | P | Dogger Bank | 25/05/2018 | *Spisula subtruncata* | *C. hysoscella* (100%) |
| NICO10_p2 | P/C | Dogger Bank | 25/05/2018 | *Macta stultorum* | *C. hysoscella* (100%) |
| NICO10_p3 | P | Dogger Bank | 25/05/2018 | *Spisula subtruncata* | *C. hysoscella* (100%) |
| Cyste_1 | C | Oostende beach | 19/09/2018 | *Aequepecten opercularis* | *Phoma* sp. (Fungi, 100%) |
| Cyste_2 | C | Oostende beach | 19/09/2018 | *Spisula subtruncata* | Pyropia sp. (Rhodophyta, 95%) |
| Cyste_3 | C | Oostende beach | 19/09/2018 | *Spisula solida* | failed |
| Cyste_4 | C | Oostende beach | 19/09/2018 | *Mimachlamys varia* | failed |
| Cyste_5 | C | Oostende beach | 19/09/2018 | *Mimachlamys varia* | failed |
| Cyste_6 | C | Oostende beach | 19/09/2018 | *Spisula subtruncata* | failed |
| Cyste_7 | C | Oostende beach | 19/09/2018 | *Spisula elliptica* | *C. lamarckii* (100%) |
| Cyste_8 | C | Oostende beach | 19/09/2018 | *Donax vittatus* | Hydroides sp. (Serpulidae, 80%) |
| Cyste_9 | C | Oostende beach | 19/09/2018 | *Cerastoderma edule* | *C. hysoscella* (100%) |
| Cyste_10 | C | Oostende beach | 19/09/2018 | *Spisula subtruncata* | failed |

**Notes.**
P, Polyp; C, podocyst.

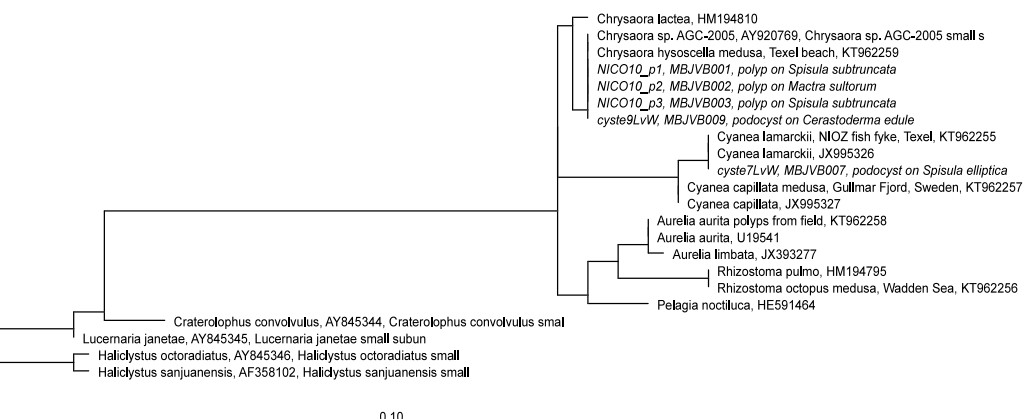

**Figure 5  RAxML tree of jellyfish reference sequences obtained from GenBank (length 1573-1585 nt), with newly obtained 18S rRNA V4–V5 sequences of polyps and podocysts (450 nt) added via ARB Parsimony.** The scale bar indicates the mean expected rates of substitutions per site. Newly added sequences are shown in italics.

identical) to the Serpulid *Hydroides* sp. and sequences of three other specimens were unreadable.

Average sizes of the two podocyst clusters that could be identified to species level were 0.22 mm ($n = 10$) for *Cyanea lamarckii* and 0.34 mm ($n = 5$) for *Chrysaora hysoscella*. These means were significantly different (one-way ANOVA, $F(1,13) = 23.51$, $p = 0.0003181$).

Podocysts were often clustered together (Fig. 4). Some clusters had a zig-zag shape with cyst size increasing, such as clusters 1, 4 and 5. Other clusters had a more random shape, such as clusters 2 and 3. The maximum number of podocysts per cluster was 11 on a shell of *Mimachlamys varia*. The average number of podocysts on a shell was 15.7, the

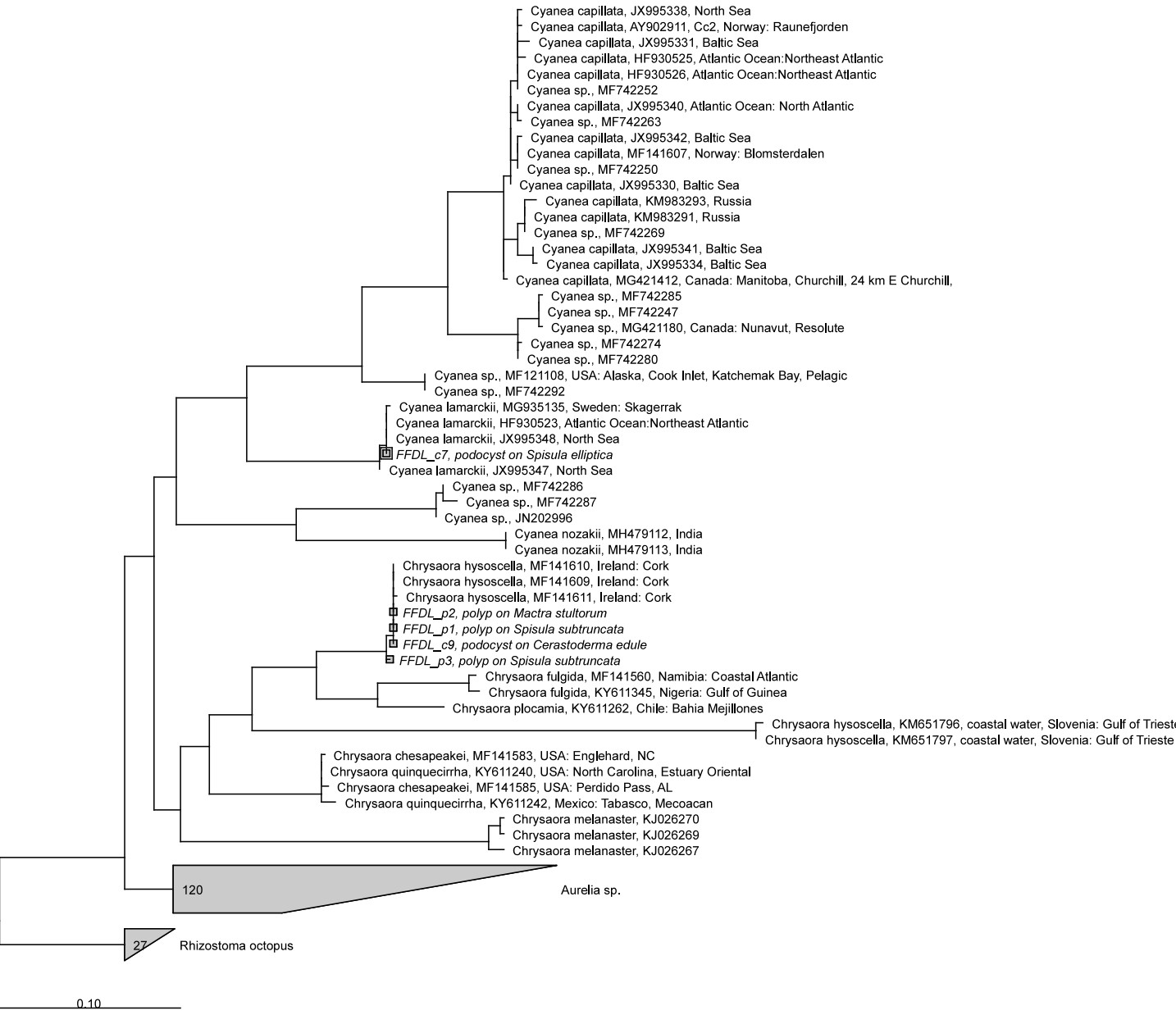

**Figure 6** RAxML tree of jellyfish reference sequences obtained from GenBank (length 470–658 bp), with newly obtained COI sequences (518–610 nt) of polyps and podocysts added via ARB Parsimony. The scale bar indicates the mean expected rates of substitutions per site. Newly added sequences are shown in italics.

minimum amount was 2 and the maximum amount was 75 podocysts on a single shell of *Mimachlamys varia*.

## DISCUSSION

Previous work based on visual inspections of artificial and natural hard substrates for scyphozoan benthic stages in the North Sea area (*Van Walraven et al., 2016*) uncovered

only polyps and podocysts of *Aurelia aurita*. Some of these polyps were also found on bivalves: living bivalves of the species *Heteranomia squamula* (one site), *Magallana gigas* (three sites) and *Mytilus edulis* (six sites). Polyp colonies sampled for this previous study were often large and conspicuous, while the podocyst and polyp clusters documented in the present study were smaller and more cryptic. This suggests that scyphozoan species other than *A. aurita* might have benthic stages that are more cryptic and occupy different niches where they are difficult to find. *Aurelia* is widely distributed and highly invasive (*Dawson, 2004*; *Bayha & Graham, 2014*), and its widespread occurrence on anthropogenic substrates documented in *Van Walraven et al. (2016)* might suggest it outcompetes other Scyphozoan species on these substrates. This might explain why for many species worldwide the benthic stages have never been observed *in situ*.

*Chrysaora hysoscella* podocysts from cultures are shown in *Morandini & Marques (2010)*, fig 34 and 35). These podocysts have a similar morphology, size and clustering as observed in our study. Similar podocyst colour and diameter are reported by *Arai (2009)* as well.

Podocysts can contribute significantly to the formation of scyphozoan jellyfish blooms. They can be dormant for more than a year, surviving conditions detrimental to polyps such as hypoxia, burial in sediment and extreme temperature or salinity (*Arai, 2009*; *Kawahara, Ohtsu & Uye, 2012*; *Thein, Ikeda & Uye, 2013*). The observation of 75 podocysts on a single shell implies that one single bivalve shell could potentially be the source of several hundred ephyrae if all podocysts on a shell developed into polyps and these would all strobilate, considering that in experiments at 10 °C *Cyanea lamarckii* polyps produced around 8 ephyrae per polyp in salinity levels encountered in the North Sea (*Holst & Jarms, 2010*).

In the North Sea, extensive oyster banks of *Ostrea edulis* occurred, which were decimated in the late 19th and early 20th century by oyster harvesting and bottom trawling (*Gercken & Schmidt, 2014*; *Houziaux et al., 2011*; *Olsen, 1883*). These oyster banks might have offered extensive habitat for scyphozoan polyps. The density of living non-reef forming bivalves can still be extremely high in the southern North Sea area. As an example, *Spisula subtruncata* has been observed in densities of several thousand ind m$^{-2}$ in the coastal Dutch North Sea (*De Bruyne et al., 2013*; *Perdon et al., 2018*). The bivalve shells on which our polyps and podocysts were observed belong to species that are common in the southern North Sea/English Channel area (*De Bruyne et al., 2013*; *Degraer et al., 2006*; *Perdon et al., 2018*). Bivalve shells can thus be an important settling substrate for Scyphozoa in the North Sea area.

Biogenic, secondary hard substrates such as dead bivalve shells are found in freshwater and in brackish and marine ecosystems worldwide, as are organisms that colonise these, which are called "skeletobionts", "skeletophytes" for plants and "skeletozoans" for animals. Bivalves been observed to form important habitat islands for hard substrate colonisers in recent as well as in fossil soft sediments, as early as the Palaeozoic (*Taylor & Wilson, 2003*). Bivalves and fossilised traces of Scyphozoan jellyfish appear in the fossil record around the same time, in the early Cambrian (*Hagadorn, Dott & Damrow, 2002*; *Cartwright et al., 2007*) making it likely that many jellyfish species have evolved to use bivalve shells as substrate for their benthic stages.

Transportation of bivalves and other hard substrates could be an important vector of introduction of invasive species of jellyfish, as has been found for the hydroid *Gonionemus vertens* (*Govindarajan & Carman, 2016*). Detection of jellyfish benthic stages using barcoding can increase our understanding of their dispersal and spread, for example through early detection of invasions or by tracing back jellyfish blooms to their location of origin.

## CONCLUSIONS

This study documents the first in situ observations of benthic stages of two scyphozoan species, *Chrysaora hysoscella* and *Cyanea lamarckii*, obtained from benthic dregdge samples and beaches. We show that podocysts are resilient enough to be detected in bivalve shells washed up on beaches, and that these washed up podocysts can still yield DNA for genetic species identification. Empty bivalve shells may also offer suitable settling substrate for other species of scyphozoa for which the benthic stage has not yet been found *in situ*. The preferred method to collect shells with identifiable podocysts or polyps on them would be to directly collect shells from the sediment surface, either by scuba diving or by collecting shells from the surface of sediment cores and subsequent preservation in >80% ethanol. The methods documented here could be applied worldwide and may help to fill the knowledge gap on scyphozoan benthic stages that still exists for many species.

## ACKNOWLEDGEMENTS

We would like to thank Marco Faasse for sharing his observations and insights on the possible whereabouts of scyphozoan polyps. We also thank the crew of RV Pelagia for their dedicated assistance during sampling and Rob Witbaard for organising and leading the NICO10 expedition leg. Hans Malschaert was valuable as Linux helpdesk. Finally, we thank Maartje Brouwer, Sanne Vreugdenhil and Kevin Sarelse for performing the lab work. Carolina Olguin Jacobson and two anonymous reviewers provided constructive and detailed comments that improved the manuscript.

### Funding

Sampling for this work was funded by the Dutch Research Council (NWO) as part of the programme "NICO: Netherlands Initiative Changing Oceans". The funders had no role in study design, data collection and analysis, decision to publish, or preparation of the manuscript.

### Grant Disclosures

The following grant information was disclosed by the authors:
The Dutch Research Council (NWO) as part of the programme "NICO: Netherlands Initiative Changing Oceans".

## Competing Interests

The authors declare there are no competing interests.

## Author Contributions

- Lodewijk van Walraven conceived and designed the experiments, performed the experiments, analyzed the data, prepared figures and/or tables, authored or reviewed drafts of the paper, and approved the final draft.
- Judith van Bleijswijk performed the experiments, analyzed the data, prepared figures and/or tables, authored or reviewed drafts of the paper, and approved the final draft.
- Henk W. van der Veer conceived and designed the experiments, authored or reviewed drafts of the paper, and approved the final draft.

## Field Study Permissions

The following information was supplied relating to field study approvals (i.e., approving body and any reference numbers):

Sampling in UK waters was approved by the Maritime Policy Unit (Legal Directorate) of the Foreign and Commonwealth Office (ref 33/2018).

## DNA Deposition

The following information was supplied regarding the deposition of DNA sequences:

Sequences are available at GenBank: MT074030–MT074034 (COI) and MT075505–MT075509 (18S).

## Data Availability

The data are available in the Supplemental Files and the sequences for COI and 18S are available at GenBank.

## Supplemental Information

Supplemental information for this article can be found online at http://dx.doi.org/10.7717/peerj.9260#supplemental-information.

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
