# Peer review of "Here are the polyps: in situ observations of jellyfish polyps and podocysts on bivalve shells"

_PeerJ, doi:10.7717/peerj.9260_

## Round 0.1 · original submission · Minor Revisions

The referees like your paper. and make good suggestions for improvements. In particular, figures must be EASY to read if printed out on paper, to people with colour blindness, and reduced in size to less than a page (as is normal in journals).

·

Basic reporting

The structure of the manuscript is hard to follow considering the findings of it. The introduction should avoid giving information that is not relevant to the findings of the manuscript. Some paragraphs need to improve the flow of ideas
The figures need small changes

Experimental design

The knowledge gap is understandable but it could be clearer
The molecular methods could be more clear, following previous studies (e.g. van Walraven et al 2016)

Validity of the findings

No comment

Additional comments

The manuscript submitted by the authors titled “Here are the polyps: in situ observations of jellyfish polyps and podocysts on bivalve shells” is very attractive in terms of the relevance of the findings. The life cycle of jellyfish has several stages, one of those stages, the polyp stage, is so small that has been extremely difficult to find in the field. Few papers have recorded the presence of polyps in the wild (Makabe et al 2014; Shibata et al 2015; van Walraven et al 2016) yet on artificial structures such as marinas, wrecks, harbours, plastic bottles and aluminium cans. To my knowledge, only Toyokawa (2011) reported finding Chrysaora pacifica polyps on shells and stones on the southern coast of Japan. Given the importance of the polyp stage since it produces the medusa stage, their finding in the wild together with the molecular methodology used in this manuscript to identify the jellyfish species stands out for the Cnidarian audience.
However, the work should be improved if considered for publication. The structure of the manuscript is hard to follow giving information that is not adding to the manuscript findings. In my understanding, the main contribution of this work is to report the discovery of polyps in the wild and the techniques applied to identify them on a species level.
In this sense, paragraphs about environmental factors that influence the reproduction of the polyps seem out of context. In my opinion, a background that describe the life cycle of jellyfish, that highlights the importance of polyps and podocysts, along with the difficulty to identify them without molecular techniques (which then can be link to the ones used in the manuscript) and state who else have found the polyps in the wild will be a better approach for the introduction. That way they can connect these main topics in their discussion with their findings.

Reviewer 2 ·

Basic reporting

The text is clear and unambiguous, and for most parts a professional English is used throughout.
Relevant literature references are included, and sufficient field background/context has been provided.
Figures and tables are OK and appropriate raw data made available.
Self-contained with relevant results to hypotheses.
The manuscript is ‘self-contained,’ and do represent an appropriate ‘unit of publication’. It includes all results relevant to the conclusion.

The research question is well defined, relevant & meaningful. In the ms it is stated how this research fills an identified knowledge gap.
The manuscript clearly defines the research question, which is relevant and meaningful.

Experimental design

Most methods are described with sufficient detail & information to replicate, but additional information is needed for the phylogenetic analyses (as indicated in the pdf).

Validity of the findings

The manuscript reports on a strategy with which we can better our understanding of jellyfish benthic stages and their ecology. I suggest that the authors expand the discussion with a paragraph on how this can contribute also to our understanding on the dispersal and spread of invasive jelly fish species such as the hydroid Gonionemus sp. This means of transportation has already been suggested but there are no references in the literature where actual polyp identifications using barcoding (or any other means) has been done.
The data on which the conclusions are based are provided and made available in an acceptable discipline-specific repository, ie Genbank.

Additional comments

The manuscript is a short (but sweet) account of how molecular technology and barcoding can be used to map and identify benthic stages of jellyfish. Some of the species has previously not been identified as benthic stages in the wild.

Annotated reviews are not available for download in order to protect the identity of reviewers who chose to remain anonymous.

Reviewer 3 ·

Basic reporting

Appropriate

Experimental design

Not applicable.

Validity of the findings

The findings are novel and well documented.

Additional comments

The manuscript “Here are the polyps: in situ observations of jellyfish polyps and podocysts on bivalve shells” by Lodewijk van Walraven, Judith van Bleijswijk and Henk W van der Veer presents first time confirmation of asexual polyp stages of two large scyphozoan jellyfish species in the North Sea.
The manuscript is well structured, written and the analyses seem appropriate. Sampling permits are explicitly stated and raw data have been deposited. I only have some minor comments outlined below:
Line 68 add at before various
Line 151 states that 2 µl DNA extract were used for PCR reactions, while in line 115 in the methods it is specified that 5µl DNA template has been used. Please check this.
Fig. 5 is not readable, I would suggest to change the light blue to another colour. The same applies to Fig. 6.
Table 1: Please incude % identity information into the table and add position data (latitude and longitude). The failed samples have been mentioned in the text and I would recommend to remove those here. Also, considering the low amount of data presented and the large number of display items, I would suggest to remove the transect figure and rather include the latitude and longitude information in Table 1.

---

## Round 0.2 · accepted · Accept

Thank you for the thorough revisions of the paper and clear response. Please note that Fig 5 and 6 have very small text so make sure that this will be large enough to be legible in the final paper.